# Exploiting the Potential of Powdered Blends of Recovered Sunflower Seed Cake Phenolics and Whey—Development of Sustainable Food Additives

**DOI:** 10.3390/foods13101433

**Published:** 2024-05-07

**Authors:** Anna Michalska-Ciechanowska, Jessica Brzezowska, Krzysztof Lech, Klaudia Masztalerz, Malgorzata Korzeniowska, Aleksandra Zambrowicz, Marek Szoltysik

**Affiliations:** 1Department of Fruit, Vegetable and Plant Nutraceutical Technology, Faculty of Biotechnology and Food Science, Wrocław University of Environmental and Life Sciences, Chełmońskiego 37, 51-630 Wrocław, Poland; jessica.brzezowska@upwr.edu.pl; 2Institute of Agricultural Engineering, Faculty of Life Sciences and Technology, Wrocław University of Environmental and Life Sciences, Chełmońskiego 37a, 51-630 Wrocław, Poland; krzysztof.lech@upwr.edu.pl (K.L.); klaudia.masztalerz@upwr.edu.pl (K.M.); 3Department of Functional Food Product Development, Faculty of Biotechnology and Food Science, Wrocław University of Environmental and Life Science, Chełmońskiego 37, 51-630 Wrocław, Poland; malgorzata.korzeniowska@upwr.edu.pl (M.K.); aleksandra.zambrowicz@upwr.edu.pl (A.Z.); marek.szoltysik@upwr.edu.pl (M.S.)

**Keywords:** sunflower seed cake, whey, spray drying, novel di-blends, sensory analysis, food reformulation

## Abstract

The management of side streams from the food industry, especially oil and dairy by-products, has become an important issue linked to the European Commission’s recommendations for a circular economy. This study aimed to obtain sustainable food additives in the form of soluble-type powders composed of whey and recovered phenolics originating from sunflower seed cake. In order to valorise these di-blend products, the powders were characterised in terms of their physical, chemical, and sensory attributes. Based on the study findings, the addition of sunflower seed cake washouts (SSCWs) to whey (Wh) decreased the dry matter in the feed that affected the viscosity and drying yield. The addition of SSCWs did not have a significant effect on the physical properties of powders, except for colour. By-product management proposed in the study resulted in the production of nutritious and ready-to-use products in powder form with improved functional properties in terms of phenolic compounds and antioxidant capacity. The powders were sensorially appealing with a tangy sourness entwined with a delicate interplay of sweet and salty flavours, which can be easily incorporated into different types of foodstuffs.

## 1. Introduction

Sunflower seed cake (SSC) is a by-product of oil production, which can be performed by different methods that moderate the final chemical composition of the product [1]. Such alterations are an important factor, especially when the further application of SSC is concerned. The SSC has been recognised as a valuable feed component for a variety of livestock. However, more recently, due to the global need for sustainable protein sources, the recognition of SSC as a plant food source for humans has increased [2]. Proteins are the main constituents extracted from sunflower seed cake accounting for up to 20–25% of its composition. This value can further increase up to approx. 50% depending on the extraction method or by removing the oil with organic solvents [3,4,5].

In addition to protein, the SSC also contains phenolics, mainly phenolic acids that can account for up to 7% of their mass, among which the major is chlorogenic acid [6]. Their presence decreases protein solubility, which hinders the production of isolates and concentrates from sunflower seed cake [7].

In order to fully exploit the potential of this by-product, numerous attempts have been made to improve the yield of protein extraction, including de-phenolisation of SSC [4]. Although phenolic removal has been shown to be more effective in organic solvents–water mixtures, the green extraction techniques using only water are increasingly practised due to their lower environmental impact [8]. Consequently, such a pre-treated SSC is safe and can be used for further steps of protein extraction. Furthermore, the latest research is being carried out as part of the FERBLEND project (https://ferblend.webspace.tu-dresden.de, accessed on 21 January 2024) on the utilisation of SSC in new platform products that can potentially be used for human consumption, among others, through fermentation processes (Figure 1) [9]. However, the sunflower seed cake phenolic-rich aqueous washouts (SSCWs) generated during this procedure remain unexploited as the next by-product of SSC management. 

One way to manage these SSCWs is to reintroduce them to the food chain by conversion into powders, which is in line with the policy of circular agriculture, and consequently increase the sustainability of food production [10,11]. The powder production can be performed by different means, but only a few enable the conversion of liquefied raw materials into fine powder. One of them is spray drying, recognised as an advantageous strategy in the food industry as it is simultaneously a time- and cost-saving process with low energy consumption compared to conventional drying techniques [11]. In addition, the quality of the final product is often comparable or superior to lyophilised products [12]. However, due to the liquid form of the substrates and their chemical composition, which is often unfavourable for the drying process (high content of low-molecular-weight compounds), this process in most cases requires the use of a high-molecular-weight carrier to ensure its feasibility and satisfactory product yield. 

Driven by current trends, the search for more sustainable carriers compared to those commonly used has also begun in the field of spray drying, and among the numerous food by-products that have been tested for encapsulation to date, whey appears to be a promising one [13]. In addition to emulsifiers, fat replacers, or gelling agents, whey proteins in the form of concentrates, isolates, or hydrolysates have been successfully investigated for their encapsulating properties during spray drying [13,14,15]. Whey (Wh), which can be classified into sweet and acid, is a side stream of cheese-making containing valuable protein fractions, including β-lactoglobulin, α-lactalbumin, immunoglobulins, and bovine serum albumin, which, besides offering beneficial biological functions, also display satisfactory techno-functional features [16]. 

In the literature on this subject, there is growing interest in searching for mutual complementarity between plant-sourced foods and animal-sourced foods. Consistent with this, Jiang et al. found that chlorogenic acid–whey protein isolate interaction enhanced the techno-functional properties of the protein, as well as that under simulated digestion conditions, this phenolic compound promoted the breakdown of protein into smaller peptides, which displayed synergistic to chlorogenic acid antioxidant activity [17]. Therefore, the combination of phenolic acid-rich washouts with whey seems to be a promising approach that fits this concept. 

Although the development of environmentally friendly technology has made the transformation of food industry by-products into high-quality powders to be feasible, the paramount factor determining the demand for such a product is the sensory aspect. Therefore, another reason driving the concept of combining phenolics with proteins in the design of new food products is the phenomenon of interaction that occurs between the aforementioned components, which can positively affect the final quality of the product, including sensory properties, to a different extent [18]. Chung et al. pointed out that the complexation of anthocyanins with the whey protein isolate can stabilise phenolic colourants [19], while the whey proteins, mainly β-lactoglobulin, have been shown to reduce the astringent taste of wine, being at the same time a more selective treatment agent for polymeric phenolics than commercially used gelatine during winemaking [20]. In terms of real food, phenolic acid–protein interactions have been tested in bread, for example [21], but due to the complexity and variety of matrices, knowledge regarding this remains limited, especially for highly processed powdered products originating from different sources (animal- and plant-based ones). Furthermore, as mentioned by Altin et al., the existing literature is still relatively sparse in addressing the impact of processing on the nature and direction of these interactions [18]. According to Shahidi and Pan, the processing technologies applied can affect the processed matrix at several levels, determining the subsequent fate of the individual components (degradation, release, transformation, interactions, etc.), and thus the final quality of the product, which can be enhanced or depleted [22]. For this reason, case-specific research is needed to identify matrix–processing–quality interrelationships, which are fundamental especially when considering the development of novel food products for human consumption. 

Taking this into account, the aim of this study was to recognise if the phenolic-rich aqueous washouts of sunflower seed cake, can be valorised by fusion with whey, serving as a carrier for spray drying when converted into a powder form. Additionally, this study examined the effect of spray drying di-blend by-products on the sensory properties of the powders produced. Consistently, our hypothesis stated that incorporating the phenolic-rich aqueous washouts with relatively low dry matter content into whey can protect the thermolabile compounds, mainly phenolics from the influence of thermal processing during drying, and may alter the sensory properties of the powders obtained. This type of natural and sustainable product could be easily incorporated as an additive to enrich other food commodities, e.g., athletes’ nutritional supplements and instant food products, particularly with selected phenolics and amino acids. Such materials, as well as the management of these products, are within the reach of both industrial companies and local producers, who can facilitate the valorisation of these by-products into high-quality commodities. In return for covering the expense of the waste disposal, there is an opportunity to benefit from it.

## 2. Materials and Methods

### 2.1. Materials

The whey (Wh) was obtained after the production of Dutch-type renneted maturing cheeses. The whey was separated after 40 min of enzymatic coagulation using natural calf haemolysin and a cheese curd process (stirring at a coagulation temperature of 32 °C) followed by gradual reheating (5 °C/min) to a final temperature of 39 °C. After the process, the grain was separated from the whey (co-product), which was destined for powder production. 

Sunflower seed cake (SSC) was obtained by cold pressing from a material consisting of 70% whole sunflower seed and 30% hulled sunflower seed (Złoto Polskie s.c., Kalisz, Poland). Before further processing, the material was ground in a sieve with an aperture size of 1 mm (Retsch SM 2000, Haan, Germany). The removal of phenolics from SSC (de-phenolisation) [6], was made by washing 100 g of grounded SSC with 7800 g of cold tap water (17 °C) for 15 min (established experimentally). After washing, the material was separated by a cloth filter and the aqueous SSC washouts (SSCWs) were collected for further experiment. 

For powder production, the feed formulations composed of sunflower seed cake washouts (SSCWs) and whey (Wh) were mixed (Table 1) and submitted to spray drying. 

### 2.2. Methods

#### Spray Drying

The feed was spray-dried using an APV ANHYDRO LAB1 spray dryer (Østmarken, Søborg, Copenhagen, Denmark) with an air inlet temperature of 180 ± 2 °C and an average air outlet temperature of 78 ± 4 °C. Spray drying parameters were adjusted to obtain an outlet temperature below 80 °C in order to minimise the denaturation of whey protein denaturation [23]. The feed pump flow rate was 3 L/h, the nozzle air pressure was 0.15 MPa, and the aspiration rate was 100%. The powder samples were vacuum-packed and stored at −20 °C until analysis.

### 2.3. Physical Properties

#### 2.3.1. Viscosity

The (dynamic) viscosity of the liquid feed was determined using a Vibro Viscometer SV-10 (A&D Company, Ltd., Tokyo, Japan). The temperature of liquid feed was 25 ± 1 °C. The measurements were performed in triplicates (*n* = 3).

#### 2.3.2. Dry Matter

Dry matter (dm) of the liquid feed and the powder collected was measured (*n* = 3) by vacuum drying, using a vacuum oven (Vacucell 111 EcoLine MMM Group, Munchen, Germany) with a vacuum pump (MZ 2C NT + AK + EK, Vacuum Brand GmbH, Wertheim, Germany) at a temperature of 80 ± 1 °C at a pressure of 300 Pa for 24 h [24].

#### 2.3.3. Product Yield

Product yield (powder recovery) was determined as the ratio of the dry matter of powder collected after drying to the weight of total solids in the feed. The results are presented in percentages (*n* = 2) [25].

#### 2.3.4. Water Activity

The water activity of powder was measured (*n* = 3) using the water activity meter AquaLab DewPoint 4TE (Decagon Devices Inc., Pullman, WA, USA) at 25 ± 0.5 °C.

#### 2.3.5. Bulk Density, True Density, Porosity

The bulk density (*ρb*) of the powders was determined as the ratio of their mass (*m*) to their bulk volume (*V_b_*). The powders were weighed (XA 60/220/X Radwag; Radom, Poland), and the bulk volume was measured with a graduated cylinder (10 ± 0.5 mL).
(1)$$\rho_{b}=\frac{m}{V_{b}}$$

The true density (*ρt*) was determined as the ratio of the mass (*m*) of powder to the volume of this powder using the gas pycnometer (*V_t_*) HumiPyc II (InstruQuest Inc., Boca Raton, FL, USA), using argon under 220 kPa.
(2)$$\rho_{t}=\frac{m}{V_{t}}$$

The porosity (*ε*) of the powders was calculated according to Equation (3) described by Michalska et al. [26].
(3)$$\varepsilon =(1-\frac{\rho_{b}}{\rho_{t}})\cdot 100$$

#### 2.3.6. Solubility

The solubility of the powders was determined in duplicate (*n* = 2) by measuring a weight difference (%) according to Cano-Chauca et al. [27]. 

#### 2.3.7. Colour

The colour of powders was determined (*n* = 5) using a Minolta Chroma Meter CR-400 (Minolta Co., Ltd., Osaka, Japan) with reference to the colour space, CIE *L*a*b**. The total change in colour (*dE*) was calculated following the equation described by Saikia et al. [28]. Chroma (*C**) and hue angle (*h*) of powders were calculated according to the procedure described by Jedlińska et al. [29]. The powder obtained from the drying of the whey was used as a target (Wh 100%). 

### 2.4. Chemical Properties

#### 2.4.1. Total Phenolic Content by the Folin–Ciocalteu Method

The total phenolic content (TPC) was determined using the Folin–Ciocalteu colorimetric method [30]. Extracts from powders were prepared by dissolving 50 mg of the sample in 1.7 mL of an 80% aqueous methanol. The liquid feeds were measured directly with an appropriate dilution. The mixture was manually mixed and then subjected to sonication for 15 min. After 24 h extraction at 4 °C and re-sonication for 15 min, the samples were centrifuged (19,000× *g*, 20 °C; MPW-251, MPW Med. Instruments, Warsaw, Poland). The absorption measurement was carried out using the Synergy H1 spectrophotometer (BioTek Instruments Inc., Winooski, VM, USA) at a wavelength of λ = 750 nm. The results were reported as mg of gallic acid equivalent (GAE) per 100 g of dm based on duplicate measurements (*n* = 2).

#### 2.4.2. Total Phenolic Content by Fast Blue BB method

To omit interferences caused by non-phenolic antioxidants, reducing agents, and proteins that may contain them, a method using Fast Blue BB reagent was performed according to Medina [31], which is a competitive and more selective procedure than the Folin–Ciocalteu assay. For this purpose, 1 mL of methanolic powders extract (prepared as described above) was combined with 100 μL of 0.1% Fast Blue BB reagent made in the same solvent in which the extracts were prepared, followed by the addition of 100 μL of 5% NaOH. Each sample was supplemented with a blank sample analysis representing 1 mL of extract to which 200 μL of deionised water was added to measure interferences caused by non-phenolic compounds. The samples were mixed and then left in a shaded area for 1 h. After this time, 200 μL of each sample and blank was transferred, and the absorbance was measured at λ = 420 nm. The results were expressed as mg of GAE per 100 g sample dm (*n* = 2).

#### 2.4.3. Antioxidant Capacity In Vitro

The in vitro antioxidant capacity of liquid feeds and powders’ extract was assessed spectrophotometrically using TEAC ABTS^•+^ and FRAP assays (Synergy H1, BioTek Instruments Inc., USA) according to Michalska-Ciechanowska et al. [12]. The results were presented as mmol Trolox equivalent per 100 g of dm (*n* = 2). 

#### 2.4.4. Soluble Tryptophan

The liquid feeds prepared as described above (Table 1) and aqueous extracts of powders (0.2 g per 12 mL of water; *w/v*) were tested in terms of the level of soluble tryptophan [32]. For this purpose, 200 µL of each sample (*n* = 2) was transferred into 96-well black plate, and the fluorescence intensity was measured at the excitation and emission wavelength: λ_ex_ = 290 and λ_em_ = 340 nm, using the Synergy H1 spectrophotometer (BioTek Instruments Inc., USA). The results were reported in arbitrary units (AU). 

#### 2.4.5. Determination of Available Amino Groups by *o*-phthaldialdehyde Method

The extracts (*n* = 2) of the powders prepared (30 mg/10 mL) in 6% SDS solution were extracted for 30 min (sonicated and vortexed each 10 min) to obtain a homogeneous suspension. The liquid feeds were directed to the subsequent step without any additional manipulation. The samples were then filtered through Whatman No. 40 paper filters and subjected to further analysis as described by Michalska et al. [32], adapted to the microplate method. To 50 µL of the sample, 100 µL of distilled water was added, followed by 100 µL of OPA reagent (consisting of 16.4 mg of *o*-phthaldialdehyde dissolved in 2.5 mL of 95% ethanol, 25 mL of borate buffer (0.1 M; pH = 9.5), 400 µL of 10% β-2-mercaptoethanol solution, 5 mL of 20% SDS solution, and made up to 100 mL with distilled water in a volumetric flask). The blank consisted of 50 µL of sample and 200 µL of distilled water. The fluorescence was measured at λ_ex_ = 340 and λ_em_ = 455 nm using the Synergy H1 spectrophotometer (BioTek Instruments Inc., USA). A calibration curve was established using *N*^α^-acetyl-*L*-lysine (10–250 µM) as an external standard, and the results were calculated in g of *N*^α^-acetyl-*L*-lysine per 100 g of wb (wet basis) or dm (*n* = 2).

#### 2.4.6. Sodium Dodecyl Sulphate–Polyacrylamide Gel Electrophoresis (SDS–PAGE) Analysis

Sodium Dodecyl Sulphate–Polyacrylamide Gel Electrophoresis (SDS–PAGE) analysis was performed according to Laemmli [33]. The samples were diluted with the buffer (including SDS and β-mercaptoethanol as the reducing reagents) and were denatured. Then, the samples were loaded (10 µL) onto gel slabs (10%). At the end of the analysis, the gel slabs were stained with Coomassie Brilliant G-250 dye. Protein molecular weights were analysed by Infinity Capt (BioRad, Hercules, CA, USA) program for electropherogram analysis.

### 2.5. Sensory Analysis

Sensory analysis of powdered samples was conducted using the 9-point hedonic scale rating according to ISO 11136:2014 Sensory analysis—Methodology—General guidance [34]. Within this scale, the highest score of 9 was assigned to ‘like extremely’, while the lowest score of 1 was designated for ‘dislike extremely’. A group of 11 participants (4 men and 7 women) aged 25–49, consisting of postgraduate students and faculty members from the Faculty of Biotechnology and Food Science at Wrocław University of Environmental and Life Sciences (UPWr, Wrocław, Poland), was assembled to assess selected quality attributes in terms of liking (appearance, colour, smell, taste, texture, mouthfeel, stickiness, as well as overall acceptability) and perceived intensity (sourness, bitterness, sweetness, saltiness, off-flavour, plant taste, earthy taste) of the powders. The samples were tested in the laboratory at room temperature and randomly served in disposable plastic cups labelled with 3-digit codes. All sensory analysis procedures were approved by the UPWr Rector’s ethical commission (decision number NON00000.0011.4.2024 signed on 15 January 2024). 

### 2.6. Statistical Analysis

The data collected were statistically analysed using the STATISTICA 13 programme (StatSoft, Tulsa, OK, USA). Significant differences (*p* < 0.05) between samples were determined by applying the HSD Tukey test, while the relationships between specific variables were assessed by calculating the Pearson correlation coefficient (*r*). Sensory analysis data were analysed by ANOVA applying the Duncan test for the determination of significant differences between sample average values at *p* < 0.05. 

## 3. Results and Discussion

### 3.1. Physical Properties

The SSCWs had a very low dry matter content and were therefore not spray-dried. Whey was set as a control sample in the study. Several feed compositions were proposed as presented in Table 1, and the results of the physical parameters of these feeds are presented in Table 2. The addition of SSCWs to whey decreased dry matter in the feed, which can be connected to its low dm that simply diluted the feed. The viscosity of the whey was found to be around 1.33 mPa·s, which is typical for dairy products with this amount of dry matter and at this temperature [35]. SSCW100% had a viscosity of 0.940 mPa·s, which is similar to the viscosity of water at the same temperature and further supports the low concentration of this material. 

Both parameters, namely dry matter and viscosity, affect the product yield. Adding 10% of SSCWs to whey resulted in a 7% decrease in the viscosity of liquid feed. Furthermore, an increase in SSCWs in the composition did not lead to the decrease in this parameter, which can be explained by the interactions between proteins and phenolics that led to the creation of particles with higher molecular mass [36], increasing the viscosity of the liquid feed. Continued addition of SSCWs (above 40%) led to a significant reduction in viscosity. When considering the product yield, it can be noticed that the addition of SSCWs facilitated the spray drying process and consequently increased the product yield; however, this increase was not statistically significant. Similar product yield in laboratory conditions were previously obtained in studies on the spray drying of insect protein-polyphenol particles [37].

Regarding the physical parameters of powders (Table 3), the dry matter was above 97.32%, which is typical for spray-dried products in powder form, and the values above 95% of dm are considered optimal to minimise agglomeration and microbial contamination [37]. The addition of SSCWs did not have a significant effect on the dry matter of the powder but increased the product yield. The higher share of SSCWs in the composition resulted in a powder with a lower value of water activity. The small differences may be due to the changes in dry matter and water availability. Similar results were obtained during the spray drying of beetroot juice [29]. It is worth noting that the whey contained 17.6 times more dry matter compared to the SSCWs; therefore, the addition of the 50% of SSCWs (based on volume) introduced only 5.4% of its dry matter. Overall, the water activity for all powders was below 0.2, which indicates the safe value of *a_w_* that can limit the growth of microorganisms [38]. No significant differences were reported for both true density and bulk density, but the addition of SSCWs resulted in a slight increase in true density. However, significant differences can be reported in the case of porosity. A lower share of whey in the composition resulted in a higher porosity of powders. It can be associated with a slight increase in true density and a simultaneous decrease in bulk density, which resulted in a significantly higher value of porosity. The variant with the highest share of SSCWs had significantly higher porosity. Powders were also characterised in terms of solubility that is an important parameter of the further utilisation of powders, i.e., in addition to other food products. The solubility of spray-dried powders was in the range of powdered whey protein products [39]. The incorporation of 20% (*v*/*v*) of sunflower seed cake washouts into whey resulted in slightly better solubility of the products; however, each successive dose increase of 10% did not statistically significantly affect this attribute. Previously, Rawel et al. [39] stated that the addition of selected phenolic acids to whey reduced the solubility of products obtained that was dependent on the specific components as well as on the pH of the solution. In the case of the study performed, the increase in solubility may be a result of the dilution effect of whey dry basis as the addition of 50% of SSCWs result in the inclusion of 5.4% dry matter to the solution. 

The colour of the powders was evaluated, and the results are presented in Table 4. The addition of SSCWs decreased the value of *L**. The darkening of the sample can be associated with the dark colour of SSCWs. Negative values of coordinate *a** indicate that the colour of the powders was closer to the green. The values of *b** and *C** changed in the same way. Chroma determines the degree of saturation of colour, and lower values indicate that the colour is more diluted and changed towards grey after the addition of SSCWs [29]. The hue angle indicates how much the colour is different from its primary value (red, blue, green, and yellow). A hue angle ≤ 90° means yellow, while ≤180° means green [40]. The values obtained in this study increased with the addition of SSCWs but remained close to 100°. Positive values of *b** decreased towards a bluish hue; however, they still remained positive. Similarly, the addition of SSCWs influenced the total colour change (*dE*), which was to be expected as this parameter was calculated in relation to pure whey powder. The *dE* value above 3 means that the colour change is very distinct, and *dE* < 1.5 is not distinguishable by the human eye [41]. In the case of Wh5/SSCW5, the value was 7.09 ± 0.06, which means that it can be easily spotted as different from the whey sample.

### 3.2. Chemical Properties

The main focus of the study was to verify the possibility of obtaining a powdered form of whey-based products with the addition of sunflower seed cake washouts as one of the strategies applied for the management of dairy and oil by-products into an easy-to-handle form of powder. Taking into account that the SSCWs constituted only up to 5.4% of dry matter in di-blended liquid feeds (in the case of 50% of Wh and 50% of SSCWs) submitted to spray drying, the study examined how the addition of sunflower seed cake aqueous washouts in different proportions modify the total phenolic content, antioxidant capacity, as well as changes in the protein profile of powders obtained. 

### 3.3. Total Phenolic Content and Antioxidant Capacity

Among the samples analysed, the content of total phenolic compounds in the control, i.e., whey (no addition of SSCWs), was the lowest; however, the TPC values were not linked to the presence of phenolics, but to the content of selected amino acids having the ability to reduce the Folin–Ciocalteu reagent [42]. This is due to the colour changes that may occur during this spectrophotometric method that allows for the measurement of reducing substances and may not necessarily be represented by phenolic components [43]. The addition of SSCWs in a quantity of more than 10% (*v*/*v*) resulted in a statistically significant increase in TPC values that was even 2.8-fold higher when 50% of SSCWs was added in comparison to Wh (control) (Table 5). This was related to the presence of phenolics, mainly phenolic acids originating from SSC washouts that are a source of relatively strong antioxidants, such as, chlorogenic, neochlorogenic, and caffeic acids, among others [6].

Interestingly, as mentioned before, the addition of 50% (*v*/*v*) of SSCWs to whey resulted in the inclusion of only 5.4% of dry matter in the whey solution, confirming its strong potential for phenolic enrichment in obtained products. 

In the case of antioxidant capacity measured by the ability of samples to scavenge the ABTS^•+^ radical cations, whey itself also showed the antioxidant properties which can be linked to the presence of specific amino acids, especially those with acidic side residues, which include glutamic and aspartic acid [44]. Furthermore, the antioxidant capacity of whey can be modified depending on the processing methods, including the obtainment of whey protein products. Mann et al. proved that hydrolysates of whey protein had strong antioxidant properties that were enhanced by the type of whey protein isolate hydrolysation [45]. These properties were attributed to hydrophobic and aromatic amino acids present in whey concentrates. The addition of sunflower seed cake washouts significantly affected the antioxidant capacity of the analysed samples. Incorporation of 50% of SSCWs into whey resulted in a 3.6-fold increased ability to scavenge ABTS^•+^ radicals compared to 10% SSCWs’ added products. A similar observation was made for the FRAP method (Table 5) as both methods were based on the similar mechanism of the reaction [46].

In the case of powders, the content of total phenolics determined by the Folin–Ciocalteu method ranged from 24.4 to 126.3 mg GAE/100 g dm for, respectively, whey powder and a formulation consisting of 50% whey and 50% SSC washouts (*v*/*v*) (Table 5). Similarly to liquid feeds, the addition of washouts to whey resulted in a significant increase in the TPC content in powders produced by spray drying, even up to 5-fold, when the addition of 50% of washouts to the composition was made (Table 5). Thongzai et al. proved that the alterations in the total phenolic content in samples composed of whey protein with selected phenolic acids were associated with the type and concentration of these compounds [47]. The phenolic components were the major contributors to the antioxidant capacity of powders as TEAC ABTS and FRAP values were highly correlated with TPC (*r* = 0.9832 and *r* = 0.9985, respectively). What is more, in the current study, the content of TPC and antioxidant capacity values in the liquid feed calculated per dry matter were significantly higher when compared to those obtained for powders. As proteins are well known for their ability to change the surface tension of solutions [48], coating formulations based on whey may be able to keep the phenolics locked in their microcapsules [49]. This may also affect the extractability of phenolics from the powders during sample preparation. 

Taking into account the interference of reducing substances, including proteins on the TPC values, the Fast Blue BB assay was used for a deeper examination of their content in analysed powders. Similarly to Pico et al. [50], a higher sensitivity of Fast Blue BB assay was confirmed when compared to TPC by the Folin–Ciocalteu method; however, a considerable variation in phenolic level and pattern amongst the samples was observed.

According to this method, powders in which washouts constituted from 10 up to 30% (*v*/*v*) of the formulation did not show statistically significant differences between the content of phenolics (Figure 2). The addition of more than 40% of SSC washouts led to the obtainment of products with a significantly higher TPC content (more than double) when compared to the rest of the mixes that were used for spray drying. Thus, the addition of less than 40% of SSCWs to whey led to the obtainment of products with a similar total phenolic content in powders.

### 3.4. Tryptophan Fluorescence Intensity

The tryptophan fluorescence intensity (TFI) was used to monitor the alterations in liquid feed compositions and powders. Tryptophan is considered as one of the amino acids that do not produce a green-coloured derivative while reacting with oxidised dimers of chlorogenic acids [51]. In the study, the tryptophan fluorescence intensity of the whey sample exhibited the strongest fluorescence emission at 340 nm during an excitation of 290 nm. The TFI values were the highest for the whey samples in both forms, i.e., liquid feed and in the powder (no SSCWs addition), and it decreased within the addition of SSCWs that was linked to the altered content of this component in analysed mixes (Figure 3). Previously, Cao et al. indicated that the fluorescence intensity of tryptophan decreased within the increasing concentration of gallic acid and epigallocatechin gallate in whey protein concentrate solution at pH = 7, pointing out that both compounds may modify whey protein structure and functional properties, but to a different extent [52].

In the case of liquid feeds, the 50% addition of SSCWs resulted in a 2.6-fold lower TFI value compared to its counterpart with a 10% (*v*/*v*) addition. When powders were considered, the decrease in TFI was less (1.9 times lower). This may be due to the stabilisation of the product caused by spray drying as reported by Ajmera and Scherließ [53]. Thus, processing by spray drying of novel di-blend compositions could be a promising approach towards the production of a more functional product.

### 3.5. Available Amino Groups by o-phthaldialdehyde Method

To further investigate the di-blends composed of side streams from the oil and dairy industries, the *o*-phthaldialdehyde (OPA) method was used to monitor the available amino groups in liquid and powder forms. In the case of liquid feeds (before spray drying), the amino group calculated per wet basis was the highest in whey and decreased linearly with the addition of SSCWs (Figure 4A) as a result of the dilution effect. However, the addition of 50% (*v*/*v*) of SSCWs caused only a 33% decrease in available amino group content, pointing to the functional properties of di-blends that have at the same time a strong antioxidant potential (Table 2).

Taking into account the prepared powders, the level of available amino groups was considerably higher (approx. 4 times) than in the case of liquid feeds when calculated per dry matter. This phenomenon can be attributed to (1) the densification of the matrix by the removal of water, as well as (2) the relatively mild drying conditions (exposure to high temperatures of spray drying, but for a very short time), which cause fewer alterations than other heat treatments, including protein modifications [53]. Interestingly, only for the Wh5/SSCW5 sample the lower content of available amino groups compared to whey (control) was observed. In turn, the Wh7/SSCW3 was characterised by the highest available amino group content. Given the presumed composition of the sunflower seed cake washout matrix, which is likely to contain not only phenolic acids but also other constituents, including the water-soluble proteins extracted from the sunflower seed cake during the aqueous washing process used, this could probably be attributed to this explanation [54]. Moreover, it is worth noting that the powders did not follow the same pattern as their respective liquid feeds before drying, which proves the processing–matrix composition interrelation effect on the ultimate properties of the product. 

### 3.6. Sodium Dodecyl Sulphate–Polyacrylamide Gel Electrophoresis (SDS–PAGE) Analysis

For the SDS-PAGE analysis, a molecular weight marker (10–250 kDa) was applied to estimate the molecular weight of whey and whey–sunflower seed cake washouts di-blends in powder form. Figure 5 shows whey composed of ɑ-lactalbumin, ɑ-lactoglobulin, and bovine serum albumin (based on the literature data). It was indicated that the product composed of whey and sunflower seed cake washouts contained ɑ-lactalbumin and ɑ-lactoglobulin as major components with a molecular weight of, respectively, 14.2 kDa and 18 kDa. On the basis of the SDS-PAGE analysis, it was shown that within the increase in the content of SSCWs in di-blends in powder form (up to 50%, *v*/*v*), the presence of both components slightly decreased. Another significant contributor of proteins present in the mixed compositions was a protein with a molecular weight greater than 50 kDa and lower than 75 kDa—bovine serum albumin (66.3 kDa). 

On the basis of the electrophoretic analysis, it was shown that the addition of 40% and 50% of sunflower seed cake washouts (*v*/*v*) resulted in the appearance of bands with a molecular weight between 25 and 37 kDa that was linked to the weaker intensity of those with a molecular weight 14.2 kDa and 18 kDa. Thus, the stronger formation of molecular fraction between 25 and 37 kDa was noticed, pointing to the possibility of binding by phenolics present in washouts with whey proteins that was confirmed by an increased intensity of these bands. It was probably linked to an increase in molecular weight of the conjugates formed [39,55]. What is more, it was indicated that the binding was not effective in the case of a sample composed of whey and SSCWs added below 80% (*v*/*v*). This may also be linked with the content of total phenolic components as determined by the Fast Blue BB method, indicating the binding of those constituents to whey proteins [56].

### 3.7. Sensory Analysis of the Powders

The results of the sensory analysis of whey and phenolic-rich sunflower seed cake aqueous washout powders’ attractiveness, carried out by the trained panellists (Table 6), showed no significant (*p* > 0.05) differences between samples in all of the analysed sensory indicators, except colour. The di-blend that was composed of 60% whey and 40% washouts (Wh6/SSCW4) scored the lowest (4.82 points) compared to the one with 80% of whey with a complementary 20% of SSCWs (6.18 points). The powders were characterised by average high scores (higher than 5 on a scale of 9 points) for appearance, smell, and texture, and only slightly lower scores were given when evaluating the taste of the powders. The overall acceptability of the new dried formulations based on whey and washouts were in the middle of the 1–9-point scale, i.e., an average between 4.82 points for Wh8/SSCW2 and 5.18 points for Wh5/SSCW5 and Wh9/SSCW1. 

The whey powder mixes with phenolic-rich sunflower seed cake aqueous washouts (Wh5/SSCW5) were characterised by an average intensity of sour taste, i.e., 4.55 point out of 9, and a clear trend to higher sourness was noted along with increased whey share in the powders (Table 7). A similar trend was also observed for powders’ sweetness; however, the intensity of the trait was much lower and did not exceed 4.18 points when the highest whey concentration (90%) was used for the production of the powder. Table salt presence was indicated clearly in all analysed samples (3.91–4.36 points). For all the traits analysed that were related to the off-flavours and off-tastes, as well as bitterness, the panellists indicated a low intensity (3.73 point) in the whey powder mixes with phenolic-rich sunflower seed cake aqueous washouts (Wh5/SSCW5), with a decreasing trend along with a higher concentration of SSCWs. 

## 4. Conclusions

In this study, the phenolic-rich sunflower seed cake aqueous washouts as a by-product of sunflower seed cake de-phenolisation were successfully converted by spray drying into powders with the addition of whey, leading to the obtainment of easy-to-handle sustainable products. The addition of SSCWs facilitated the spray drying process of di-blends and consequently affected the product yield; however, the increase in this parameter was not significant. The addition of SSCWs to whey did not have a significant effect on the physical properties of powders, except for the colour, which exhibited a higher total colour difference when mixed with SSCWs. The results of the total phenolics, antioxidant capacity, available amino groups, soluble tryptophan, and SDS-PAGE analyses confirmed that the transformation of whey to SSCW-added new blend formulations in powder form led to the obtainment of products with improved functional properties. At the same time, the co-creation of novel di-blended products resulted in the formation of sensory attractive powders characterised by distinctive sourness with a pleasant glimpse of sweet and salty taste. With all the above considerations in mind, the new approach towards the addition of phenolic-rich aqueous washouts that make up by-products from the sunflower seed cake management can be a promising material for the production of high-quality food powders with broad application potential. 

## Figures and Tables

**Figure 1 foods-13-01433-f001:**
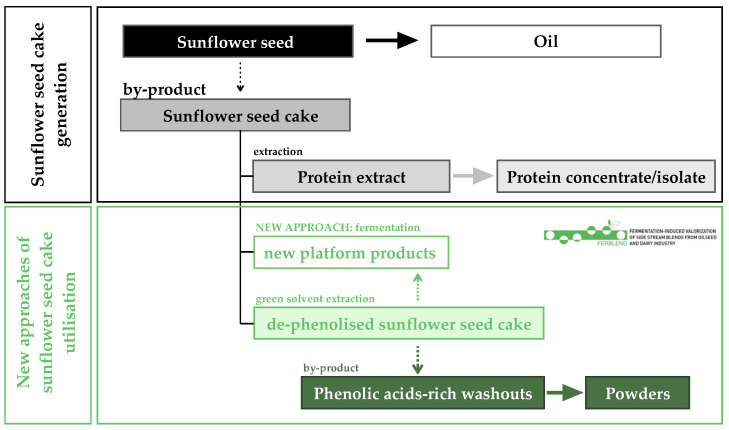
Sunflower seed cake generation scheme and novel approach towards its possible application as a food component in the frame of the FERBLEND project.

**Figure 2 foods-13-01433-f002:**
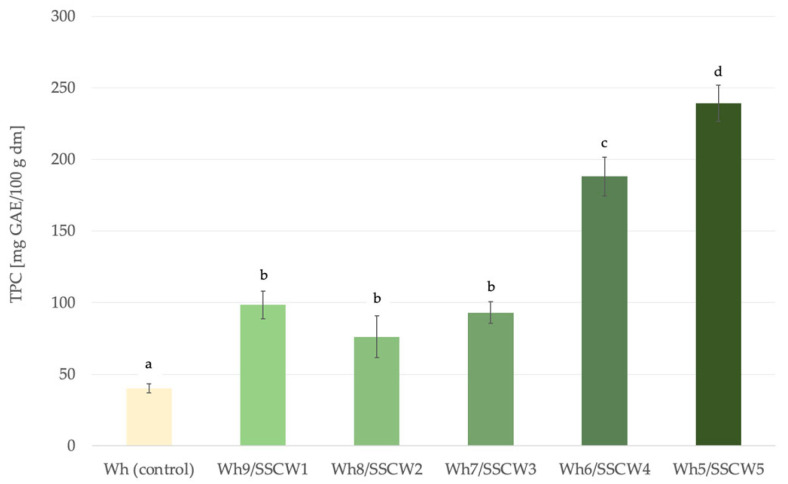
The total phenolic content by FAST Blue BB method in powders. Wh—whey; SSCW—sunflower seed cake washout; GAE—gallic acid equivalent; ^a–d^—different letters representing different samples indicate significant differences (ANOVA, HSD Tukey, *p* < 0.05).

**Figure 3 foods-13-01433-f003:**
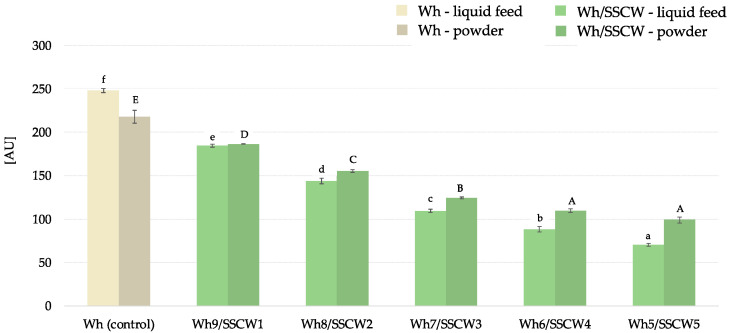
Tryptophan fluorescence intensity [AU] of liquid feeds and powders. AU—arbitrary units; Wh—whey; SSCW—sunflower seed cake washout; ^a–f^—different letters within group (liquid feeds) representing different samples indicate significant differences (ANOVA, HSD Tukey, *p* < 0.05); ^A–E^—different letters within group (powders) representing different samples indicate significant differences (ANOVA, HSD Tukey, *p* < 0.05).

**Figure 4 foods-13-01433-f004:**
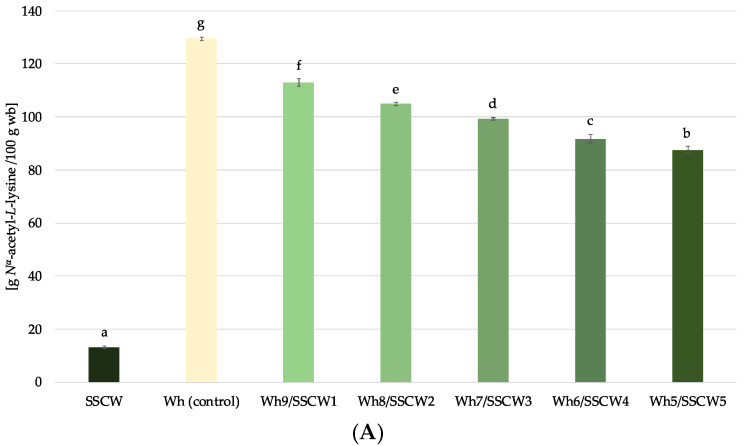
The available amino group content (*N*^α^-acetyl-*L*-lysine) in liquid feeds calculated per wet basis (**A**) and in liquid feeds and powders calculated per dry matter (**B**); Wh—whey; SSCW—sunflower seed cake washout; ^a–g^—different letters within group (liquid feeds) representing different samples indicate significant differences (ANOVA, HSD Tukey, *p* < 0.05); ^A–C^—different letters within group (powders) representing different samples indicate significant differences (ANOVA, HSD Tukey, *p* < 0.05).

**Figure 5 foods-13-01433-f005:**
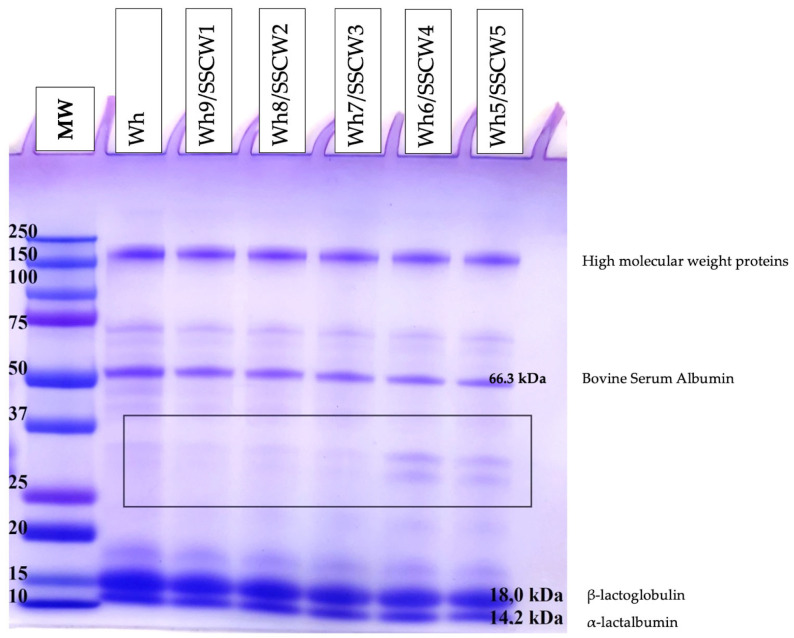
SDS-PAGE electropherogram (10% gel) of powders obtained from whey proteins and di-blends composed of whey and sunflower seed cake washouts. MW—molecular weight marker (10–250 kDa); Wh—whey; SSCW—sunflower seed cake washout; Wh9/SSCW1—whey 90% and SSCW 10%, Wh8/SSCW2—whey 80% and SSCW 20%, Wh7/SSCW3—whey 70% and SSCW 30%, Wh6/SSCW4—whey 60% and SSCW 40%, Wh5/SSCW5—whey 50% and SSCW 50%.

**Table 1 foods-13-01433-t001:** The composition of liquid feeds composed of whey (Wh) and sunflower seed cake washouts (SSCW) prepared for spray drying.

Sample	Wh (%; *v*/*v*)	SSCW (%; *v*/*v*)
Wh (control)	100	0
Wh9/SSCW1	90	10
Wh8/SSCW2	80	20
Wh7/SSCW3	70	30
Wh6/SSCW4	60	40
Wh5/SSCW5	50	50

**Table 2 foods-13-01433-t002:** Dry matter, viscosity of liquid feed compositions made from di-blends (before spray drying), and product yield after spray drying.

Sample	Dry Matter [%]	Viscosity [mPa·s]	Product Yield [%]
SSCW100%	0.420 ± 0.090 ^a^*	0.940 ± 0.014 ^a^	-
Wh (control)	7.379 ± 0.007 ^e^	1.330 ± 0.014 ^d^	39.36 ± 0.23 ^a^
Wh9/SSCW1	6.852 ± 0.251 ^de^	1.240 ± 0.014 ^c^	38.29 ± 1.40 ^a^
Wh8/SSCW2	5.867 ± 0.137 ^cd^	1.290 ± 0.016 ^cd^	40.66 ± 0.95 ^a^
Wh7/SSCW3	5.516 ± 0.053 ^cd^	1.315 ± 0.021 ^e^	40.12 ± 0.39 ^a^
Wh6/SSCW4	4.972 ± 0.288 ^bc^	1.105 ± 0.021 ^b^	41.72 ± 2.42 ^a^
Wh5/SSCW5	3.845 ± 0.233 ^b^	1.090 ± 0.014 ^b^	43.51 ± 2.63 ^a^

Wh—whey; SSCW—sunflower seed cake washout; * values followed by the same letter, within the same column, were not significantly different (*p* > 0.05) according to Tukey’s HSD test.

**Table 3 foods-13-01433-t003:** Physical properties of powders.

Sample	Dry Matter [%]	Water Activity [–]	True Density [g/cm^3^]	Bulk Density [g/cm^3^]	Porosity [%]	Solubility [%]
Wh (control)	97.55 ± 0.07 ^a^ *	0.176 ± 0.001 ^c^	1.378 ± 0.004 ^a^	0.656 ± 0.076 ^a^	52.4 ± 4.5 ^a^	78.41 ± 0.4 ^a^
Wh9/SSCW1	98.08 ± 0.28 ^a^	0.183 ± 0.007 ^d^	1.380 ± 0.003 ^a^	0.605 ± 0.043 ^a^	56.2 ± 2.5 ^ab^	80.69 ± 1.2 ^ab^
Wh8/SSCW2	98.02 ± 0.10 ^a^	0.152 ± 0.004 ^bc^	1.404 ± 0.042 ^a^	0.632 ± 0.027 ^a^	55.0 ± 1.9 ^ab^	81.75 ± 0.7 ^b^
Wh7/SSCW3	97.41 ± 0.30 ^a^	0.165 ± 0.002 ^c^	1.400 ± 0.017 ^a^	0.585 ± 0.001 ^a^	58.3 ± 0.4 ^bc^	82.43 ± 1.1 ^b^
Wh6/SSCW4	97.69 ± 0.17 ^a^	0.144 ± 0.011 ^b^	1.395 ± 0.009 ^a^	0.651 ± 0.041 ^a^	53.3 ± 2.4 ^ab^	82.46 ± 1.0 ^b^
Wh5/SSCW5	97.32 ± 0.21 ^a^	0.127 ± 0.014 ^a^	1.413 ± 0.005 ^a^	0.539 ± 0.003 ^a^	61.9 ± 0.2 ^c^	83.17 ± 0.9 ^b^

Wh—whey; SSCW—sunflower seed cake washout; * values followed by the same letter, within the same column, were not significantly different (*p* > 0.05) according to Tukey’s HSD test.

**Table 4 foods-13-01433-t004:** Colour of powders obtained from di-blended liquid feeds.

Sample	*L* (D65)*	*a* (D65)*	*b* (D65)*	*C* (D65)*	*H (D65)*	*dE*
Wh (control)	90.81 ± 0.19 ^f^ *	−3.56 ± 0.05 ^b^	16.32 ± 0.29 ^d^	16.70 ± 0.30 ^d^	102.33 ± 0.05 ^c^	-
Wh9/SSCW1	90.01 ± 0.03 ^e^	−2.95 ± 0.06 ^d^	15.16 ± 0.23 ^b^	15.45 ± 0.24 ^b^	101.03 ± 0.05 ^a^	1.54 ± 0.19 ^a^
Wh8/SSCW2	88.84 ± 0.05 ^d^	−3.04 ± 0.03 ^a^	15.26 ± 0.09 ^b^	15.56 ± 0.09 ^b^	101.27 ± 0.07 ^b^	2.30 ± 0.09 ^b^
Wh7/SSCW3	88.15 ± 0.03 ^c^	−3.10 ± 0.03 ^a^	12.58 ± 0.19 ^a^	12.95 ± 0.20 ^a^	103.84 ± 0.05 ^f^	4.62 ± 0.18 ^c^
Wh6/SSCW4	86.22 ± 0.20 ^b^	−3.21 ± 0.04 ^c^	13.38 ± 0.10 ^c^	13.76 ± 0.10 ^c^	103.51 ± 0.08 ^d^	5.47 ± 0.22 ^d^
Wh5/SSCW5	84.82 ± 0.03 ^a^	−3.07 ± 0.01 ^a^	12.57 ± 0.06 ^a^	12.94 ± 0.06 ^a^	103.71 ± 0.04 ^e^	7.09 ± 0.06 ^e^

Wh—whey; SSCW—sunflower seed cake washout; * values followed by the same letter, within the same column, were not significantly different (*p* > 0.05) according to Tukey’s HSD test.

**Table 5 foods-13-01433-t005:** The total phenolic content (TPC) by the Folin–Ciocalteu method, antioxidant capacity (TEAC ABTS and FRAP) in di-blended liquid feeds (before spray drying) and respective powders.

		TPC (Folin–Ciocalteu) [mg GAE/100 g dm]	TEAC ABTS [mmol Trolox/100 g dm]	FRAP [mmol Trolox/100 g dm]
Liquid feeds	Wh (control)	75.48 ± 6.56 ^a^	0.11 ± 0.01 ^a^	0.15 ± 0.01 ^a^
	Wh9/SSCW1	88.75 ± 3.60 ^b^	0.33 ± 0.05 ^b^	0.37 ± 0.01 ^b^
	Wh8/SSCW2	113.33 ± 1.51 ^c^	0.50 ± 0.04 ^c^	0.62 ± 0.08 ^c^
	Wh7/SSCW3	132.31 ± 1.38 ^d^	0.62 ± 0.03 ^cd^	0.72 ± 0.00 ^c^
	Wh6/SSCW4	158.85 ± 0.13 ^e^	0.76 ± 0.06 ^d^	0.90 ± 0.01 ^d^
	Wh5/SSCW5	213.05 ± 1.66 ^f^	1.19 ± 0.04 ^e^	1.28 ± 0.02 ^e^
Powders	Wh (control)	24.39 ± 0.85 ^a^	0.20 ± 0.02 ^a^	0.17 ± 0.00 ^a^
	Wh9/SSCW1	40.56 ± 2.88 ^b^	0.42 ± 0.07 ^ab^	0.27 ± 0.02 ^b^
	Wh8/SSCW2	51.77 ± 1.11 ^c^	0.59 ± 0.02 ^bc^	0.38 ± 0.00 ^c^
	Wh7/SSCW3	73.90 ± 1.45 ^d^	0.72 ± 0.04 ^cd^	0.52 ± 0.02 ^d^
	Wh6/SSCW4	96.37 ± 3.06 ^e^	0.89 ± 0.05 ^de^	0.65 ± 0.02 ^e^
	Wh5/SSCW5	126.29 ± 1.65 ^f^	1.08 ± 0.10 ^e^	0.88 ± 0.00 ^f^

Wh—whey; SSCW—sunflower seed cake washout; TPC—total phenolic content; TEAC ABTS—Trolox equivalent antioxidant capacity by ABTS; FRAP—ferric-reducing antioxidant potential; GAE—gallic acid equivalent; ^a–f^—different letters within groups in the columns: liquid feeds and respective powders representing different samples indicate significant differences (ANOVA, HSD Tukey, *p* < 0.05).

**Table 6 foods-13-01433-t006:** Sensory attractiveness of the whey powder mixes combined with phenolic-rich sunflower seed cake aqueous washouts.

Sample	Appearance	Colour	Smell	Taste	Texture	Mouthfeel	Stickiness	Overall Acceptability
Wh9/SSCW1	5.64 ± 1.29	6.00 ± 1.26	5.27 ± 1.10	4.73 ± 1.62	5.36 ± 1.80	5.36 ± 1.57	5.64 ± 1.50	5.18 ± 0.87
Wh8/SSCW2	6.00 ± 1.26	6.18 ± 1.25 *	5.18 ± 1.40	4.73 ± 1.49	5.36 ± 1.43	5.00 ± 2.10	5.36 ± 1.75	4.82 ± 1.40
Wh7/SSCW3	5.82 ± 1.33	5.64 ± 1.43	5.73 ± 1.01	4.45 ± 1.51	5.55 ± 1.63	5.09 ± 2.12	5.27 ± 1.90	5.00 ± 1.26
Wh6/SSCW4	5.00 ± 0.89	4.82 ± 0.98 *	5.18 ± 0.87	4.91 ± 2.17	6.00 ± 1.61	5.91 ± 1.76	5.36 ± 1.75	5.00 ± 1.34
Wh5/SSCW5	5.18 ± 1.60	5.09 ± 1.81	5.09 ± 1.70	4.82 ± 2.32	5.82 ± 1.72	4.45 ± 2.21	4.18 ± 2.14	5.18 ± 1.78
sem	0.18	0.19	0.16	0.24	0.22	0.26	0.25	0.18
*p*	0.33	0.11	0.77	0.98	0.86	0.53	0.38	0.97

Wh—whey; SSCW—sunflower seed cake washout; * values with asterisks are significantly different.

**Table 7 foods-13-01433-t007:** Intensity of the sensory attributes of whey powder mixes combined with phenolic-rich sunflower seed cake aqueous washouts.

Sample	Sourness	Bitterness	Sweetness	Saltiness	Off-Flavour	Plant Taste	Earthy Taste
Wh9/SSCW1	5.09 ± 1.76	2.55 ± 1.21	4.18 ± 1.47	3.91 ± 1.45	2.36 ± 1.21	1.82 ± 0.75	1.82 ± 1.17
Wh8/SSCW2	5.64 ± 1.91	2.73 ± 1.10	4.09 ± 1.64	4.36 ± 2.29	2.00 ± 0.89	1.82 ± 0.60	1.91 ± 1.14
Wh7/SSCW3	5.00 ± 1.73	3.00 ± 1.34	3.45 ± 1.37	4.00 ± 1.67	2.82 ± 1.83	2.00 ± 0.89	1.82 ± 1.08
Wh6/SSCW4	4.91 ± 1.58	3.73 ± 2.28	3.91 ± 1.58	4.09 ± 1.22	3.09 ± 1.58	2.73 ± 1.56	2.09 ± 1.04
Wh5/SSCW5	4.55 ± 1.57	3.36 ± 1.50	3.36 ± 1.75	4.36 ± 1.63	3.00 ± 1.34	2.27 ± 1.10	2.36 ± 1.36
sem	0.23	0.21	0.21	0.22	0.19	0.14	0.15
*p*	0.68	0.39	0.65	0.95	0.33	0.21	0.78

Wh—whey; SSCW—sunflower seed cake washout.

## Data Availability

The original contributions presented in the study are included in the article, further inquiries can be directed to the corresponding author.

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
