# Peer review of "Exploiting the Potential of Powdered Blends of Recovered Sunflower Seed Cake Phenolics and Whey—Development of Sustainable Food Additives"

_foods, 2024, doi:10.3390/foods13101433_

Round 1

Reviewer 1 Report

Comments and Suggestions for Authors

I suggest the following revision for your feedback:

"While the research article presents intriguing findings, I would recommend revisiting the manuscript before publication to address some minor areas for improvement:

The introduction section should be revised to include the application or product for utilizing the sunflower seed cake.

References should be provided for the spray-dry methods mentioned.

Instead of using a, b, c, d in the tables and figures, it is preferable to use a-d for clarity.

The manuscript requires language refinement, including correcting English grammar, syntax, and typographical errors throughout.

Specifically, the following changes are recommended:

Line 24 adds the before addition

Line 48 changes cake to cakes

Line 74 adds is before in line

Line 75 removes the comma after production

Line 78 removes the before food

Line 81 changes lyophilisates to lyophilisation

Line 87 changes has to have

Line 99 adds a or the before protein

Line 113 adds a before more

Line 122 adds a comma before and

Line 141 adds the before production

Line 182 adds the before liquid

Line 197 adds the before water

Line 201 adds the before bulk

Line 205 adds was before measured

Line 207 adds the before porosity

Line 221 changes was to were

Line 242 removes ‘ after powders

Line 272adds the or a before sample

Line 325 adds a comma after Further

Line 327 changes lead to led

Line 349 changes spray dried to spray-dried

Line 356 changes volume based to volume-based

Line 358 changes indicated to indicates

Line 360 changes of to in

Line 367 adds the before solubility

Line 367 changes spray dried to spray-dried

Line 369 changes increases to increase

Line 374 adds the before addition

Line 374 changes result to results

Line 374 adds the before inclusion

Line 388 adds the before colour

Line 388 adds the before powders

Line 395 adds and before yellow

Line 412 changes to obtain to of obtaining

Line 413 adds the before addition

Line 419 adds the before protein

Line 432 changes other to others

Line 432 adds a comma before and

Line 434 adds the before Folin

Line 445 removes it was

Line 459 adds A before similar

Line 460 changes the similar to a similar

Line 462 removes the comma after method

Line 466 changes was to were

Line 469 changes of antioxidant to to the antioxidant

Line 480 adds the before Folin

Line 481 changes phenolics to phenolic

Line 507 adds the before content

Line 508 adds the before addition

Line 510 adds the before mixes

Line 510 adds are before used

Line 510 adds the before addition

Line 511 changes phenolics to phenolic

Line 516 changes green coloured to green-coloured

Line 552 adds the before florescence

Line 539 changes products to product

Line 548 adds the before dilution

Line 549 adds a before 33

Line 549 changes groups to group

Line 550 adds the before functional

Line 571 changes 4-times to 4 times

Line 571 adds the before case

Line 574 changes less to fewer

Line 575 changes proteins to protein

Line 577 changes samples to the samples,

Line 578 changes with to by

Line 582 changes probable to probably

Line 583 changesto note to noting

Line 588 adds a comma after analysis

Line 590 changes indicated to indicates

Line 591 changes products to product

Line 597 adds a before molecular

Line 610 changes showed to shown

Line 611 changes seeds to seed

Line 615 changes was to were

Line 617 adds the before Fast

Line 617 changes those constituent to that cons constituent or those constituents

Line 627 adds a comma before and

Line 628 adds the  before taste

Line 629 changes was to were

Line 639 adds A before similar

Line 640 removes the comma after lower

Line 642 removes and

Line 643 changes point to points

Line 645 adds a before lowering

Line 655 changes were to was

Line 656 adds the before addition

Line 659 adds The before addition

Line 660 adds the before colour

Line 662 adds a comma before and

Line 662 adds the before transformation

Line 664 adds the before co-creation

Line 665 adds the before formation

Line 667 adds the before addition

Comments on the Quality of English Language

I suggest the following revision for your feedback:

"While the research article presents intriguing findings, I would recommend revisiting the manuscript before publication to address some minor areas for improvement:

The introduction section should be revised to include the application or product for utilizing the sunflower seed cake.

References should be provided for the spray-dry methods mentioned.

Instead of using a, b, c, d in the tables and figures, it is preferable to use a-d for clarity.

The manuscript requires language refinement, including correcting English grammar, syntax, and typographical errors throughout.

Specifically, the following changes are recommended:

Line 24 adds the before addition

Line 48 changes cake to cakes

Line 74 adds is before in line

Line 75 removes the comma after production

Line 78 removes the before food

Line 81 changes lyophilisates to lyophilisation

Line 87 changes has to have

Line 99 adds a or the before protein

Line 113 adds a before more

Line 122 adds a comma before and

Line 141 adds the before production

Line 182 adds the before liquid

Line 197 adds the before water

Line 201 adds the before bulk

Line 205 adds was before measured

Line 207 adds the before porosity

Line 221 changes was to were

Line 242 removes ‘ after powders

Line 272adds the or a before sample

Line 325 adds a comma after Further

Line 327 changes lead to led

Line 349 changes spray dried to spray-dried

Line 356 changes volume based to volume-based

Line 358 changes indicated to indicates

Line 360 changes of to in

Line 367 adds the before solubility

Line 367 changes spray dried to spray-dried

Line 369 changes increases to increase

Line 374 adds the before addition

Line 374 changes result to results

Line 374 adds the before inclusion

Line 388 adds the before colour

Line 388 adds the before powders

Line 395 adds and before yellow

Line 412 changes to obtain to of obtaining

Line 413 adds the before addition

Line 419 adds the before protein

Line 432 changes other to others

Line 432 adds a comma before and

Line 434 adds the before Folin

Line 445 removes it was

Line 459 adds A before similar

Line 460 changes the similar to a similar

Line 462 removes the comma after method

Line 466 changes was to were

Line 469 changes of antioxidant to to the antioxidant

Line 480 adds the before Folin

Line 481 changes phenolics to phenolic

Line 507 adds the before content

Line 508 adds the before addition

Line 510 adds the before mixes

Line 510 adds are before used

Line 510 adds the before addition

Line 511 changes phenolics to phenolic

Line 516 changes green coloured to green-coloured

Line 552 adds the before florescence

Line 539 changes products to product

Line 548 adds the before dilution

Line 549 adds a before 33

Line 549 changes groups to group

Line 550 adds the before functional

Line 571 changes 4-times to 4 times

Line 571 adds the before case

Line 574 changes less to fewer

Line 575 changes proteins to protein

Line 577 changes samples to the samples,

Line 578 changes with to by

Line 582 changes probable to probably

Line 583 changesto note to noting

Line 588 adds a comma after analysis

Line 590 changes indicated to indicates

Line 591 changes products to product

Line 597 adds a before molecular

Line 610 changes showed to shown

Line 611 changes seeds to seed

Line 615 changes was to were

Line 617 adds the before Fast

Line 617 changes those constituent to that cons constituent or those constituents

Line 627 adds a comma before and

Line 628 adds the  before taste

Line 629 changes was to were

Line 639 adds A before similar

Line 640 removes the comma after lower

Line 642 removes and

Line 643 changes point to points

Line 645 adds a before lowering

Line 655 changes were to was

Line 656 adds the before addition

Line 659 adds The before addition

Line 660 adds the before colour

Line 662 adds a comma before and

Line 662 adds the before transformation

Line 664 adds the before co-creation

Line 665 adds the before formation

Line 667 adds the before addition

Author Response

I suggest the following revision for your feedback:

"While the research article presents intriguing findings, I would recommend revisiting the manuscript before publication to address some minor areas for improvement:

  • The introduction section should be revised to include the application or product for utilizing the sunflower seed cake.

The additional information about the possible application of sunflower seed cake products has been added as follows:

Line 135-138: This type of natural and sustainable product could be easily incorporated as an additive to enrich other food commodities, e.g. athletes nutritional supplements, instant food products, particularly with selected phenolics and amino acids.

  • References should be provided for the spray-dry methods mentioned.

The parameters were established experimentally during the preliminary tests, and the operating conditions were fixed based on the literature that was cited in the text: Jafari, S. M., Samborska, K. Spray Drying for the Food Industry: Unit Operations and Processing Equipment in the Food Industry Eds.; Woodhead Publishing: Cambridge, MA Kidlington, 2024

  • Instead of using a, b, c, d in the tables and figures, it is preferable to use a-d for clarity.

 It was corrected according to the suggestion.

  • The manuscript requires language refinement, including correcting English grammar, syntax, and typographical errors throughout.

The manuscript has been corrected according to the suggestion.

Specifically, the following changes are recommended:

  • Line 24 adds the before addition – It was done according to the suggestion.
  • Line 48 changes cakes to cake – It was done according to the suggestion.
  • Line 74 adds is before in line – It was done according to the suggestion.
  • Line 75 removes the comma after production – It was done according to the suggestion.
  • Line 78 removes the before food – It was done according to the suggestion.
  • Line 81 changes lyophilisates to lyophilization – It was changed into lyophilised products.
  • Line 99 adds a or the before protein – It was done according to the suggestion.
  • Line 113 adds a before more – It was done according to the suggestion.
  • Line 122 adds a comma before and – It was done according to the suggestion.
  • Line 141 adds the before production – It was done according to the suggestion.
  • Line 182 adds the before liquid – It was done according to the suggestion.
  • Line 197 adds the before water – It was done according to the suggestion.
  • Line 201 adds the before bulk – It was done according to the suggestion.
  • Line 205 adds was before measured – the word ‘measured’ has been crossed out
  • Line 207 adds the before porosity – It was done according to the suggestion.
  • Line 221 changes was to were – It was done according to the suggestion.
  • Line 242 removes ‘ after powders – It was done according to the suggestion.
  • Line 272adds the or a before sample – It was done according to the suggestion.
  • Line 325 adds a comma after Further – It was done according to the suggestion.
  • Line 349 changes spray dried to spray-dried – It was done according to the suggestion.
  • Line 356 changes volume based to volume-based – It was done according to the suggestion.
  • Line 358 changes indicated to indicates – It was done according to the suggestion.
  • Line 360 changes of to in – It was done according to the suggestion.
  • Line 367 adds the before solubility – It was done according to the suggestion.
  • Line 367 changes spray dried to spray-dried – It was done according to the suggestion.
  • Line 369 changes increases to increase – It was done according to the suggestion.
  • Line 374 adds the before addition – It was done according to the suggestion.
  • Line 374 changes result to results – It was done according to the suggestion.
  • Line 374 adds the before inclusion – It was done according to the suggestion.
  • Line 388 adds the before colour – It was done according to the suggestion.
  • Line 388 adds the before powders – It was done according to the suggestion.
  • Line 395 adds and before yellow – It was done according to the suggestion.
  • Line 412 changes to obtain to of obtaining – It was done according to the suggestion.
  • Line 413 adds the before addition – It was done according to the suggestion.
  • Line 419 adds the before protein – It was done according to the suggestion.
  • Line 432 changes other to others – It was done according to the suggestion.
  • Line 432 adds a comma before and – It was done according to the suggestion.
  • Line 434 adds the before Folin – It was done according to the suggestion.
  • Line 445 removes it was – It was done according to the suggestion.
  • Line 459 adds A before similar – It was done according to the suggestion.
  • Line 460 changes the similar to a similar – It was done according to the suggestion.
  • Line 462 removes the comma after method – It was done according to the suggestion.
  • Line 469 changes of antioxidant to to the antioxidant – It was done according to the suggestion.
  • Line 480 adds the before Folin – It was done according to the suggestion.
  • Line 481 changes phenolics to phenolic – It was done according to the suggestion.
  • Line 507 adds the before content – It was done according to the suggestion.
  • Line 508 adds the before addition – It was done according to the suggestion.
  • Line 510 adds the before mixes – It was done according to the suggestion.
  • Line 510 adds are before used – It was done according to the suggestion.
  • Line 510 adds the before addition – It was done according to the suggestion.
  • Line 511 changes phenolics to phenolic – It was done according to the suggestion.
  • Line 516 changes green coloured to green-coloured – It was done according to the suggestion.
  • Line 552 adds the before florescence – It was done according to the suggestion.
  • Line 539 changes products to product – It was done according to the suggestion.
  • Line 548 adds the before dilution – It was done according to the suggestion.
  • Line 549 adds a before 33 – It was done according to the suggestion.
  • Line 549 changes groups to group – It was done according to the suggestion.
  • Line 550 adds the before functional – It was done according to the suggestion.
  • Line 571 changes 4-times to 4 times – It was done according to the suggestion.
  • Line 571 adds the before case – It was done according to the suggestion.
  • Line 574 changes less to fewer – It was done according to the suggestion.
  • Line 575 changes proteins to protein – It was done according to the suggestion.
  • Line 577 changes samples to the samples – It was done according to the suggestion.
  • Line 578 changes with to by – It was done according to the suggestion.
  • Line 582 changes probable to probably – It was done according to the suggestion.
  • Line 583 changesto note to noting – It was done according to the suggestion.
  • Line 588 adds a comma after analysis – It was done according to the suggestion.
  • Line 590 changes indicated to indicates – It was done according to the suggestion.
  • Line 591 changes products to product – It was done according to the suggestion.
  • Line 597 adds a before molecular – It was done according to the suggestion.
  • Line 610 changes showed to shown – It was done according to the suggestion.
  • Line 611 changes seeds to seed – It was done according to the suggestion.
  • Line 617 adds the before Fast – It was done according to the suggestion.
  • Line 617 changes those constituent to that cons constituent or those constituents – It was done according to the suggestion.
  • Line 627 adds a comma before and – It was done according to the suggestion.
  • Line 628 adds the  before taste – It was done according to the suggestion.
  • Line 629 changes was to were – It was done according to the suggestion.
  • Line 639 adds A before similar – It was done according to the suggestion.
  • Line 640 removes the comma after lower – It was done according to the suggestion.
  • Line 642 removes and – It was done according to the suggestion.
  • Line 643 changes point to points – It was done according to the suggestion.
  • Line 645 adds a before lowering – It was done according to the suggestion.
  • Line 655 changes were to was – It was done according to the suggestion.
  • Line 656 adds the before addition – It was done according to the suggestion.
  • Line 659 adds The before addition – It was done according to the suggestion.
  • Line 660 adds the before colour – It was done according to the suggestion.
  • Line 662 adds a comma before and – It was done according to the suggestion.
  • Line 662 adds the before transformation – It was done according to the suggestion.
  • Line 664 adds the before co-creation – It was done according to the suggestion.
  • Line 665 adds the before formation – It was done according to the suggestion.
  • Line 667 adds the before addition – It was done according to the suggestion.

Reviewer 2 Report

Comments and Suggestions for Authors

This manuscript describes the effects of addition of sunflower seed cake washouts (SSCW) to whey on the physical and chemical properties as well as sensory attributes of the mixed feed powders. The topic is interesting but need minor revision.

1.       Inclusion of sensory test results of “Control” sample (Wh) will be helpful for readers to understand the effects of addition of Wh on the sensory quality of the feeds. Would the authors add the sensory test results of “Control” (Wh) in Tables 6-7.

2.       In the section Sodium Dodecyl Sulfate–Polyacrylamide Gel Electrophoresis (SDS–PAGE) analysis, the possible mechanism of the appearance of bands with a molecular weight between 25 and 37 kDa should be discussed in more details.  

3.       About the ABSTRACT, it looks like an introduction. Some important results might be included in the ABSTRACT.

4.       Format of references list should be checked carefully: LWT should be replaced by LWT-Food Sci. Technol.

Author Response

Comments and Suggestions for Authors

This manuscript describes the effects of addition of sunflower seed cake washouts (SSCW) to whey on the physical and chemical properties as well as sensory attributes of the mixed feed powders. The topic is interesting but need minor revision. 

  • Inclusion of sensory test results of “Control” sample (Wh) will be helpful for readers to understand the effects of addition of Wh on the sensory quality of the feeds. Would the authors add the sensory test results of “Control” (Wh) in Tables 6-7.

Control sample (sweet whey) was not included in the study because the sensory evaluation panel was trained toward the sensory descriptors on the basis of the powdered whey sample (considered as a commercial product). Whey was presented to panelists, and they were asked which traits (descriptors) related to the product were the most important for them. The main objective for the sensory analysis was to describe the new products but not the comparison of the product with the "control" standard sample.

  • In the section Sodium Dodecyl Sulfate–Polyacrylamide Gel Electrophoresis (SDS–PAGE) analysis, the possible mechanism of the appearance of bands with a molecular weight between 25 and 37 kDa should be discussed in more details.

The stronger appearance of the bands in the case of samples in which the addition of sunflower seed cake washouts was more than 80% was probably linked to the increase MW of the conjugates formed. In the rest of the analyzed sample, the degree of the conjugation could be lower, thus not so visible as in the case of the mentioned above samples. Interestingly, this also indicated that the binding capacity of whey and SSCW was not effective when less than 80% of SSCWs was added to whey (v/v). 

Line 627-629: Thus, the stronger formation of molecular fraction between 25 and 37 kDa was noticed, pointing to the possibility of binding by phenolics present in washouts with whey proteins that was confirmed by increased intensity of these bands. It was probably linked to increase in MW of the conjugates formed [39, 55]. What is more, it was indicated that the binding was not effective in the case of sample composed of whey and SSCW added below 80% (v/v). This may be also linked with the content of total phenolic components as determined by the Fast Blue BB method indicating the binding of those constituents to whey proteins [56].

  • About the ABSTRACT, it looks like an introduction. Some important results might be included in the ABSTRACT. 

It was done according to the suggestion.

  • Format of references list should be checked carefully: LWT should be replaced by LWT-Food Sci. Technol.

It was done according to the suggestion.

Reviewer 3 Report

Comments and Suggestions for Authors

The study is interesting, it could be enriched in some parts, as detailed in the comments.

Detailed comments

Line 150: Check the text “milled SCC with 7 800 g”

Spray drying: Add more information to spray drying experiments. How did you get the powders? How did you chosen the experimental parameters? Have you done any previous experiments?

Physical properties: Among the physical properties described, the authors do not consider the particle size of the powders and the powders morphology. Do the authors have this information? Do you have any information about the stability of the powders?

Porosity: Specify the pb and pt terms in the equation.

Figure 4: Improve resolution

Line 624: “Wh6/SSCW6” Check, there is a mistake.

Comments on the Quality of English Language

Minor editing of  English language required

Author Response

Detailed comments

  • Line 150: Check the text “milled SCC with 7 800 g”

I was changed according to the suggestion:

Line 150-152: The removal of phenolics from SSC (de-phenolisation) [6], was made by washing 100 g of grounded SSC with 7 800 g of cold tap water (17 °C) for 15 min (established experimentally).

  • Spray drying: Add more information to spray drying experiments. How did you get the powders? How did you chosen the experimental parameters?Have you done any previous experiments?

The experimental setup was established based on the literature data as already mentioned in the comment to Reviewer 1 (Jafari, S. M., Samborska, K. Spray Drying for the Food Industry: Unit Operations and Processing Equipment in the Food Industry Eds.; Woodhead Publishing: Cambridge, MA Kidlington, 2024) as well as based on the preliminary studies. The powders were obtained according to the description provided in the Methods section, namely: air inlet temperature of 180 ± 2 °C; an average air outlet temperature of 78 ± 4 °C; feed pump flow rate of 3 L/h; the nozzle air pressure of 0.15 MPa; and the aspiration rate 100%. The parameters were set to avoid overheating of the whey as well as ensure the highest quality of powder.

  • Physical properties: Among the physical properties described, the authors do not consider the particle size of the powders and the powders morphology. Do the authors have this information? Do you have any information about the stability of the powders?

The particle size of the powders was not considered at this point, and neither did the powder morphology and stability tests. The main aim of the study was to check the feasibility of spray drying as a method to obtain powders based on sunflower seed cake washouts and whey. Moreover, the quality tests focused on the phenolic content as well as sensory tests. However, the Authors are thankful for this comment and further physical tests will be for sure considered in the future.

  • Porosity: Specify the pb and pt terms in the equation.

Equations were added according to the suggestion.

  • Figure 4: Improve resolution

 It has been done according to the suggestion.

  • Line 624: “Wh6/SSCW6” Check, there is a mistake.

It was corrected.

Round 2

Reviewer 1 Report

Comments and Suggestions for Authors

The authors have diligently incorporated the majority of the comments and suggestions provided by the reviewers during the manuscript revision process.

Author Response

Authors are grateful for the comment.